# Synthesis of macrocyclic nucleoside antibacterials and their interactions with MraY

Takeshi Nakaya[1], Miyuki Yabe[1], Ellene H. Mashalidis[2,11], Toyotaka Sato[3,4], Kazuki Yamamoto [1,5], Yuta Hikiji[1], Akira Katsuyama [1,5,6], Motoko Shinohara [7], Yusuke Minato[7], Satoshi Takahashi[8,9], Motohiro Horiuchi[3,4], Shin-ichi Yokota [10], Seok-Yong Lee [2] ✉ & Satoshi Ichikawa [1,5,6] ✉

The development of new antibacterial drugs with different mechanisms of action is urgently needed to address antimicrobial resistance. MraY is an essential membrane enzyme required for bacterial cell wall synthesis. Sphaerimicins are naturally occurring macrocyclic nucleoside inhibitors of MraY and are considered a promising target in antibacterial discovery. However, developing sphaerimicins as antibacterials has been challenging due to their complex macrocyclic structures. In this study, we construct their characteristic macrocyclic skeleton via two key reactions. Having then determined the structure of a sphaerimicin analogue bound to MraY, we use a structure-guided approach to design simplified sphaerimicin analogues. These analogues retain potency against MraY and exhibit potent antibacterial activity against Gram-positive bacteria, including clinically isolated drug resistant strains of *S. aureus* and *E. faecium*. Our study combines synthetic chemistry, structural biology, and microbiology to provide a platform for the development of MraY inhibitors as antibacterials against drug-resistant bacteria.

Bacterial antimicrobial resistance (AMR) poses a severe threat to human health worldwide[1]. According to the recent estimates of AMR burden, it is estimated that 4.95 million deaths were associated with AMR in 2019, including 1.27 million deaths being attributable to AMR[2]. The estimates indicate that AMR is a severe health problem whose magnitude is comparable or greater than HIV and malaria. In order to address AMR, the development of new antibacterial drugs with novel mechanisms of action is urgently needed[3–6]. Choosing the class of candidate compounds as a starting point is critical for this. Of the 162

antibacterial drugs approved by FDA from 1981 to 2019, 89 are natural products and their derivatives[7], indicating the important role of natural products in antibacterial drug discovery. However, due to the chemical complexity natural products and their derivatives often makes their total chemical synthesis difficult. Therefore, designing simplified analogues that retain activity is an important objective in the medicinal chemistry of natural products.

The choice of target is important for the development of new antibacterial drugs[8]. Peptidoglycan is the main component of the

[1]Faculty of Pharmaceutical Sciences, Hokkaido University, Kita-12, Nishi-6, Kita-ku, Sapporo 060-0812, Japan. [2]Department of Biochemistry, Duke University School of Medicine, Durham, NC 27710, USA. [3]Laboratory of Veterinary Hygiene, School/Faculty of Veterinary Medicine, Hokkaido University, Kita-18, Nishi-9, Kita-ku, Sapporo 060-0818, Japan. [4]Graduate School of Infectious Diseases, Hokkaido University, Sapporo 060-0818, Japan. [5]Center for Research and Education on Drug Discovery, Faculty of Pharmaceutical Sciences, Hokkaido University, Kita-12, Nishi-6, Kita-ku, Sapporo 060-0812, Japan. [6]Global Institution for Collaborative Research and Education (GI-CoRE), Hokkaido University, Kita-12, Nishi-6, Kita-ku, Sapporo, 060-0812 Sapporo, Japan. [7]Department of Microbiology, Fujita Health University School of Medicine, 1-98 Dengakugakubo, Kutsukake-cho, Toyoake, Aichi 470-1192, Japan. [8]Division of Laboratory Medicine, Sapporo Medical University Hospital, South-1, West-16, Chuo-ku, Sapporo 060-8543, Japan. [9]Department of Infection Control and Laboratory Medicine, Sapporo Medical University School of Medicine, South-1, West-16, Chuo-ku, Sapporo 060-8543, Japan. [10]Department of Microbiology, Sapporo Medical University School of Medicine, South-1, West-17, Chuo-ku, Sapporo 060-8556, Japan. [11]Present address: Pfizer Global Research & Development, Eastern Point Road, Groton, CT 06340, USA. ✉e-mail: seok-yong.lee@duke.edu; ichikawa@pharm.hokudai.ac.jp

bacterial cell wall and is constructed by cross-linking glycan and polypeptide chains[9,10]. A membrane-anchored process in the peptidoglycan biosynthesis is known as the lipid cycle, which starts from the transfer of phospho-*N*-acetylmuramyl pentapeptide from UDP-*N*-acetylmuramyl pentapeptide (Park's nucleotide) to the phospholipid undecaprenyl phosphate ($C_{55}$-P) (Fig. 1a). The transfer reaction is catalyzed by phospho-*N*-acetylmuraminic acid (MurNAc)-pentapeptide translocase, known as MraY[11–13], which is an integral membrane enzyme. The reaction product, undecaprenyl pyrophosphoryl-*N*-acetylmuramyl pentapeptide (lipid I), is glycosylated with *N*-acetylglucosamine (GlcNAc) by MurG to form lipid II. After lipid II is flipped from the cytoplasm to the periplasm by the lipid II flippase MurJ[14,15], it is polymerized by glycosyltransferases (SEDS) and penicillin-binding proteins (PBP) to form the networked structure of peptidoglycan. In 2013, the crystal structure of MraY from *Aquifex aeolicus* (MraY$_{AA}$) was solved[16], revealing the overall architecture.

MraY is highly conserved among Gram-positive and Gram-negative bacteria, as it is essential for bacterial replication. In addition, conventional antibacterial agents do not act on MraY, which makes it an attractive target for drug-resistant bacterial drugs[17–21]. There are several classes of nucleoside natural product inhibitors targeting MraY with promising activity against pathogenic bacteria; the liposidomycins[22–24], caprazamycins[25,26], capuramycins[27,28], mureidomycins[29–31], muraymycins[32], and tunicamycins[33,34] (Supplementary Fig. 1). Each MraY inhibitor shares a uridine moiety, but they otherwise differ in their core chemical structures.

Sphaerimicins A-D are nucleoside natural products isolated from *Sphaerisporangium* sp. SANK60911 by Van Lanen and coworkers using a genome mining approach focusing on uridine-5′-aldehyde transaldolase, which catalyzes an aldol reaction with glycine derived from L-threonine and uridine-5′-aldehyde[35]. Sphaerimicin A exhibits strong MraY inhibitory activity (IC$_{50}$ 13.5 ng/mL (13.9 nM) for MraY) and

### a) lipid cycle of peptidoglycan biosynthesis

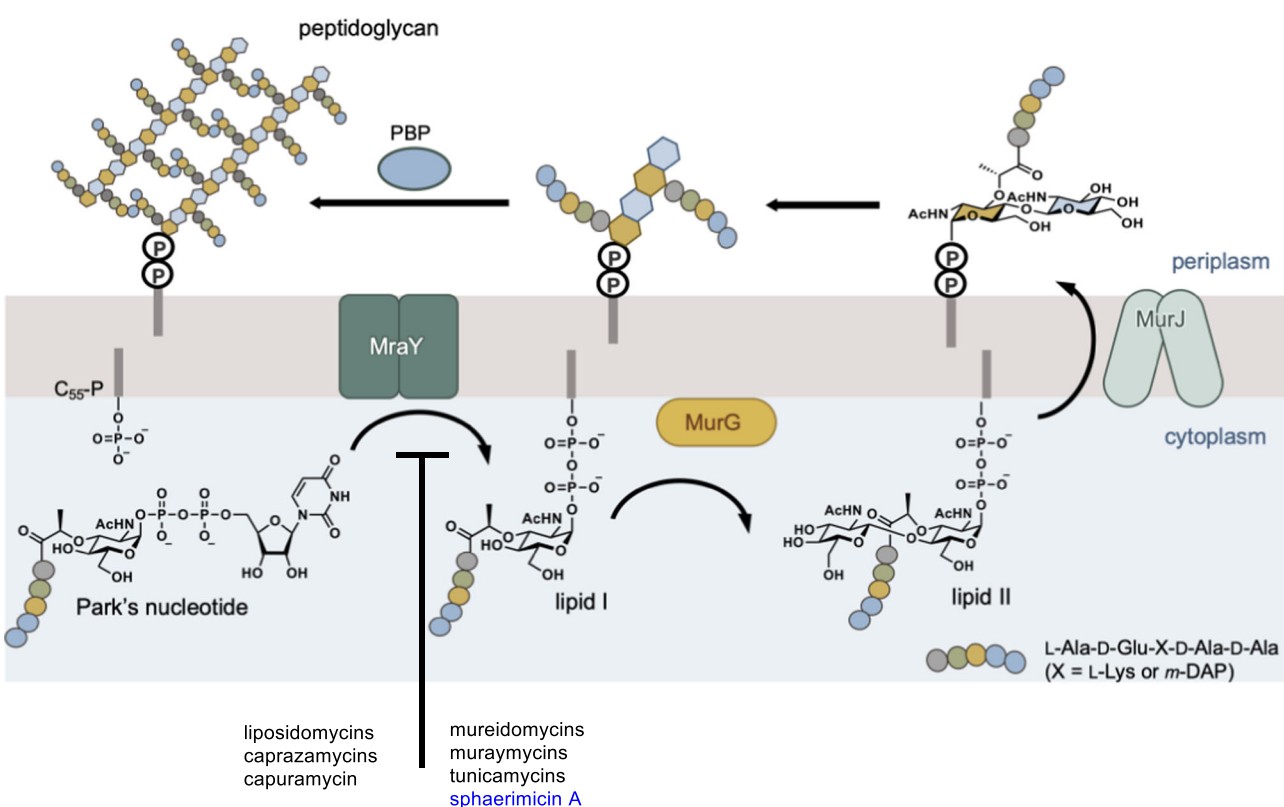

### b) structure and biological activities of sphaerimicin A

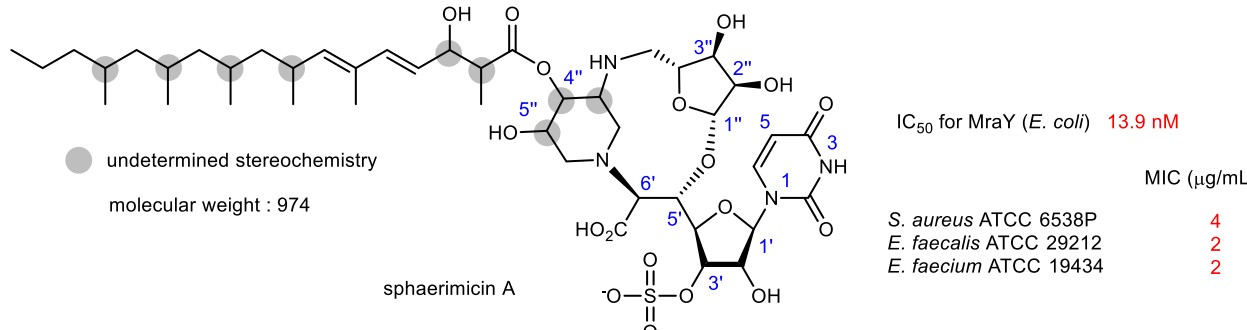

IC$_{50}$ for MraY (*E. coli*)   13.9 nM

MIC (μg/mL)

| | |
|---|---|
| *S. aureus* ATCC 6538P | 4 |
| *E. faecalis* ATCC 29212 | 2 |
| *E. faecium* ATCC 19434 | 2 |

**Fig. 1 | Peptidoglycan biosynthesis and MraY inhibitors. a** A lipid cycle of peptidoglycan biosynthesis and **b** chemical structure of sphaerimicin A. IC$_{50}$ 50% inhibitory concentration, MIC minimum inhibitory concentration, PBP penicillin-binding protein.

promising antibacterial activity against Gram-positive bacteria (MIC 1–16 μg/mL) (Fig. 1b). Sphaerimicins consist of a 5′-glycyluridine, an aminoribose, a highly substituted piperidine, and a highly methyl-branched fatty acid. Furthermore, the hydroxy group at the 3′-position of uridine is sulfated. The most intriguing chemical feature of sphaerimicins is their macrocyclic structure fused with the aminoribose and the piperidine ring, resulting in one of the most complex chemical structures among nucleoside natural products to date. The conformation of sphaerimicins induced by its unique macrocyclic structure is expected to be highly restricted, thereby the architecture of sphaerimicins is regarded as a conformationally restricted version of liposidomycins, caprazamycins and muraymycins (Supplementary Fig. 1); although their molecular interactions with MraY remain to be elucidated. Generally, a conformationally restricted molecule has an entropic advantage upon binding to the target. Using a conformationally restricted molecule as a scaffold facilitates analogue design because the conformation of newly introduced functional groups is more readily predicted. Therefore, the core chemical structure of sphaerimicins provides a potential lead scaffold for MraY inhibitors.

In this study, we design simplified sphaerimicin analogues, which we term SPMs. We not only construct the complex macrocyclic skeleton found in sphaerimicins, but we also solve the three-dimensional structure of one of these analogues, SPM-1, bound to MraY$_{AA}$. Our structural and biological analyses reveal that the stereochemistry of the macrocyclic core of the SPMs is critical for inhibitor potency. Building upon this structural information, we design a more simplified SPM analogue that retains potency against MraY and antibacterial activity.

## Results and discussion

### Molecular design of the simplified sphaerimicin analogues

The piperidine ring of the sphaerimicins is highly substituted with three contiguous stereogenic centers (3‴, 4‴, 5‴-positions), however, their relative and absolute configurations have not been determined. Preparing all possible eight stereoisomers requires substantial effort. The aim of this study is to find a stereoisomer of a sphaerimicin skeleton with potent MraY inhibitory activity. First, we conducted a molecular modeling study to predict conformations of these stereoisomers (for details, see Supplementary Information). The conformation of the all possible eight diastereomers (3‴, 4‴, 5‴) SRS-RRR were investigated by the molecular mechanics (MM) calculation and the resulting lowest conformers within 5.0 kcal/mol from the global minimums were categorized by the conformation of piperidine ring (Fig. 2a and Supplementary Fig. 2). All the global minimum structures of SRS-RRR have a piperidine ring in a chair conformation with the nitrogen heteroatom in an axial position. Based on these analyses, the first-step design of simplified sphaerimicin analogues (SPM-1 and SPM-2, Fig. 2b) was executed as follows. First, the 3‴-amino group is involved in the 11-membered ring located at the center of the bridged ring, and the stereochemistry at this position would have the most significant impact on the global minimum conformation of the molecule, falling into two classes (3‴-S: SRS, SRR, SSR, SSS vs. 3‴-R: RSR, RSS, RRS, RRR). Therefore, both diastereomers at the 3‴-amino group were selected. If the relative stereochemistry of the 3‴-amino group and the 4‴-hydroxy group are in a syn-relationship, then migration of the acyl group at the 4‴-hydroxy group to the 3‴-amino group, which is predicted to exist as a free amine at physiological pH (Supplementary Tables 1 and 2), is expected for SSR, SSS, RRS, and RRR. During the course of the structural determination of the sphaerimicin A, the fatty acyl chain was removed by basic conditions to obtain the core structure[35]. However, no acyl migration to the 3‴-amino group was reported; therefore, we hypothesized the relative stereochemistry of the 3‴-amino group and the 4‴-hydroxy group is trans configuration as in SRS, SRR, RSR, and RSS. The stereochemistry at the 5‴-hydroxy

group is not expected to influence the global conformation of the molecule (SRS vs. SRR, SSR vs. SSS, RSR vs. RSS, RRS vs. RRR); consequently, the relative stereochemistry of the 4‴, 5‴-diol was constructed with a cis configuration, a decision driven solely by the synthetic accessibility. Accordingly, we initially focused on SRS (3‴S, 4‴R, 5‴S) and RSR (3‴R, 4‴S, 5‴R) that were selected among eight possible diastereomers. Overall, the absolute stereochemistry of all three substituents on the piperidine ring is inverted between SRS and RSR. Based on our previous structure–activity relationship studies on muraymycins[36,37] and caprazamycins[38,39] we hypothesized that the branched acyl chain found in sphaerimicins can be simplified to the palmitoyl group. Lastly, the sulfate group at the 3′-hydroxy group at the uridine moiety was removed. This simplification was inspired by the observed improvement in antibacterial activity of the structurally related liposidomycins upon removal of their sulfate group at the 3″-hydroxy position[24,38]. Finally, we designed two diastereomers, SPM-1 and SPM-2 (Fig. 2b). The molecular design removes six stereogenic centers and reduces molecular weight from 974 to 784 Da compared to the original chemical structure of sphaerimicin A.

### Synthesis of the SPM-1 and SPM-2

Our retrosynthetic analysis to the synthesis of SPM-1 and SPM-2 is shown in Fig. 3. The characteristic macrocycle and the piperidine ring of A were constructed simultaneously by a double-reductive amination of aminodialdehyde B in the last stage of the synthesis. It was expected that the formation of the first 11- or 13-membered ring would be facile because both formyl groups in B can react with the amine and the second cyclization forming a 6-membered ring was predicted to be easier. Either route a or route b proceed to give the same product A, and this strategy allows the construction of two rings in a single operation. The dialdehyde B, which is a cyclization precursor, was envisioned to arise from a cyclopentene C by oxidative cleavage of the olefin. In order to access SPM-1 and SPM-2 efficiently, a cyclopentenyl group was introduced by an asymmetric Tsuji–Trost allylic alkylation reaction of D and racemic cyclopentene E[40–42]. By changing the stereochemistry of the asymmetric ligand[43–45], two stereoisomers can be efficiently synthesized from the same cyclopentene.

The synthesis of SPM-1 and SPM-2 is described in Fig. 4. The asymmetric allylic alkylation, which is the first key step to the synthesis of SPM-1 and SPM-2, was investigated. The 5′-azidoribosyluridine derivative 1[46] was transformed to a suitably protected sulfonamide 2 with a 58% yield over three steps. Its Tsuji–Trost asymmetric allylic alkylation with cyclopentene 3, which was readily prepared from a known cyclopentene[47], was investigated. The reaction of 2 and 2 equivalents of racemic 3 using 4 mol% Pd$_2$(dba)$_3$ • CHCl$_3$, 16 mol% (R,R)-DACH-phenyl Trost ligand, and Et$_3$N in THF at reflux for 2 h gave 4 in 60% yield as a mixture of diastereomers (6:1). Protection of the uridine NH group as Boc was necessary to prevent allylic alkylation at the position. The reaction using (S,S)-ligand under the same conditions gave 5 with a 73% yield as a single diastereomer. With the stereodivergent intermediates in hand, the synthesis of SPM-1 and SPM-2 was pursued via the second key step, which simultaneously constructs the characteristic macrocycle and the piperidine ring by successive reductive amination. After the secondary allylic alcohol at the cyclopentene moiety of 5 was selectively protected by the MOM group, a protecting group manipulation of 6 provided 7 with a 76% yield over three steps. Dihydroxylation of the cyclopentene moiety of 7 was first investigated by using catalytic OsO$_4$ and N-methylmorpholine-N-oxide as a re-oxidant. However, no conversion was observed, presumably because of the high steric hindrance around the alkene of 7. Ultimately, this issue was overcome by employing a harsher condition with the use of DABCO[48,49]. The reaction proceeded cleanly to give the corresponding diol, which was further oxidatively cleaved to generate a dialdehyde by NaIO$_4$. After complete consumption of the diol, the Cbz group at the N$^6$-position was removed by hydrogenolysis. The

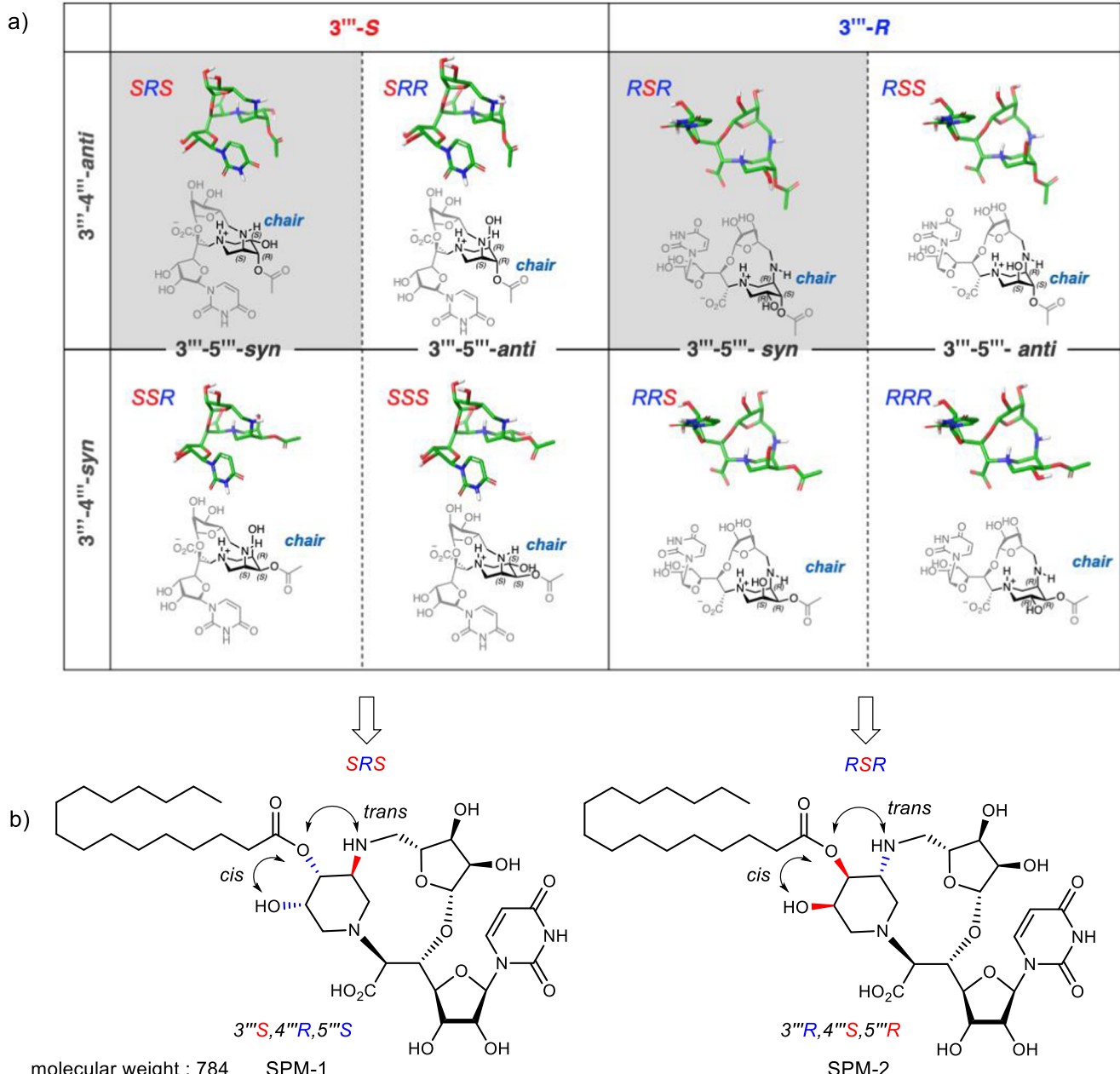

**Fig. 2 | Molecular design of simplified sphaerimicin analogues assisted by conformation analysis. a** Conformation analysis of the core structures SRS-RRR. **b** Chemical structures of simplified sphaerimicin analogues SPM-1 and SPM-2.

resulting cyclization precursor was further treated with picoline-BH$_3$[50] to promote the double-reductive amination, furnishing the desired macrocycle **8** with a 20% yield over four steps. The stereochemistry of the piperidine ring was confirmed by several NMR experiments. Namely, the ROE correlations showed that the piperidine ring has a chair-type conformation, and the proton at the 5′′′-position is coupled with the axial proton at the 6′′′ position with the *J* value of 10.4 Hz. The observed conformation is in good accordance with our initial molecular model (Fig. 2a). This stereochemical outcome also confirmed that the chemical structure of **8** and the α-positions of intermediate aldehydes are not isomerized during the reaction sequence. A mono-alkylated compound was not detected during the reaction sequence. The 4′′′-hydroxyl group of **8** was selectively acylated over the 3-NH group of the uracil ring to give **9**, and the methyl ester at the 6′-position of **9** was converted to yield the corresponding carboxylic acid **10**[51]. Finally, global deprotection of the four protecting groups of **10** with

80% aqueous TFA successfully afforded SPM-1 with a 53% yield. During the course of the deprotection, no acyl group migration was observed at all. In a manner similar to the synthesis of SPM-1, SPM-2 was prepared from **4** (Supplementary Fig. 10). The asymmetric allylic alkylation of **2** with the use of **3** with different protecting group or a diastereomer of **3** were also investigated (Supplementary Table 4). In fact, these attempts worked well and the corresponding cyclopentene derivatives were obtained with good stereoselectivity. Therefore, this divergent approach could be applied to the synthesis of other diastereomers of SPM-1 and SPM-2.

**Antibacterial activity of the SPMs**

The ESKAPE pathogens (*Enterococcus faecium*, *Staphylococcus aureus*, *Klebsiella pneumoniae*, *Acinetobacter baumannii*, *Pseudomonas aeruginosa*, and *Enterobacter* species) are the leading cause of nosocomial infections throughout the world. The antibacterial activity of SPM-1

**Fig. 3 | Retrosynthetic analysis of SPM-1 and SPM-2.** The macrocycle and the piperidine ring of A are constructed simultaneously by double-reductive amination of aminodialdehyde B. The dialdehyde B arises from a cyclopentene C by oxidative cleavage of olefin. A cyclopentenyl group is introduced by an asymmetric Tsuji–Trost allylic alkylation reaction of D and racemic cyclopentene E.

and SPM-2 was first examined against these pathogens (Table 1). SPM-1 and SPM-2 were active against Gram-positive bacteria such as *S. aureus* and *E. faecium* and were inactive against the four Gram-negative bacterial strains examined. SPMs could not penetrate into the cells owing to the outer membrane barrier, so has no activity against Gram-negative bacteria. SPM-1 exhibits antibacterial activity comparable to that of sphaerimicin A[35] with a minimum inhibitory concentration (MIC) ranging from 2 μg/mL to 4 μg/mL against the Gram-positive pathogens (Table 1). This data suggests that the branched acyl side chain of sphaerimicins can be simplified to a liner fatty acyl group and the sulfate group at the 3′-hydroxy group at the uridine moiety can be removed, as expected. By contrast, SPM-2 exhibited comparatively weaker activity with MICs increased 4-fold relative to SPM-1. When tested on *Mycobacterium* species, SPM-1 exhibited better antibacterial activity than SPM-2 against *M. tuberculosis and M. avium* with MIC values of 16 and 2 μg/mL, respectively. These results also provide insights into the correct stereochemistry of the piperidine ring of the sphaerimicins and present SPM-1 as a simplified analogue of sphaerimicins that retains potent Gram-positive antibacterial activity.

**Crystal structure of SPM-1 bound to MraY**

We previously determined the X-ray crystal structures of MraY in its apoenzyme form using a biochemically stable orthologue MraY_{AA}[16], as well as bound to natural product nucleoside inhibitors muraymycin D2[52], carbacaprazamycin (a liposidomycin analogue), capuramycin, and 3′-hydroxymureidomycin A (a ribose derivative of mureidomycin A)[53]. An X-ray crystal structure of MraY from *Clostridium bolteae* (MraY_{CB}) bound to tunicamycin is also available[54]. These five ligand-bound structures of MraY, which cover the chemical space sampled by each major class of MraY nucleoside inhibitors, collectively demonstrate that MraY inhibitors bind to the cytoplasmic face of the enzyme in a highly

conserved active site that is comprised of 34 invariant amino acid residues. The SPM-1-MraY_{AA} complex crystallized with the aid of a camelid nanobody crystallization chaperone (NB7), using an approach previously described[53]. NB7 does not interfere with MraY_{AA} activity and inhibition[53]. Although both SPM-1 and SPM-2 were screened for crystallization with MraY_{AA} and NB7, crystals were only observed for the NB7-MraY_{AA}-SPM-1 ternary complex, which diffracted to 3.65 Å. Phasing was obtained by molecular replacement and the model was refined to good geometry and statistics (Supplementary Table 6). As previously observed[16,52–54], MraY_{AA} crystallized as a dimer with one molecule of NB7 bound to each MraY_{AA} protomer on its periplasmic face. The good-quality electron density map allowed for the placement of SPM-1 (Supplementary Fig. 18). Although the assignment of macrocyclic nucleoside structure is clear, due to the limited resolution, the assignment of water molecules bound to the ligand is impossible and the exact atomic position of the aliphatic chain is unclear. The crystal structure of SPM-1 bound to MraY_{AA} demonstrates that SPM-1 recognizes a shallow binding site in MraY formed predominantly by transmembrane helices (TMs) 5, 8, 9b, and Loops C, D, and E (Fig. 5a), as is observed in the other available structures of MraY bound to its nucleoside inhibitors.

The highly conserved cytoplasmic site on the enzyme has been subdivided into six hot spots of MraY inhibition[52,53]. Each MraY inhibitor class recognizes a unique combination of hot spots, but the uridine moiety of all MraY nucleoside natural product inhibitors interact with the same site on the enzyme, termed the uridine pocket. The uridine moiety of SPM-1 also binds the uridine pocket and makes several contacts with other MraY hot spots, including sites we name the uridine-adjacent, TM9b/Loop E, hydrophobic, and Mg$^{2+}$-binding sites (Fig. 5b).

A detailed view of the SPM-1 binding site (Fig. 5c) reveals the extensive contacts the inhibitor makes with residues within each hot

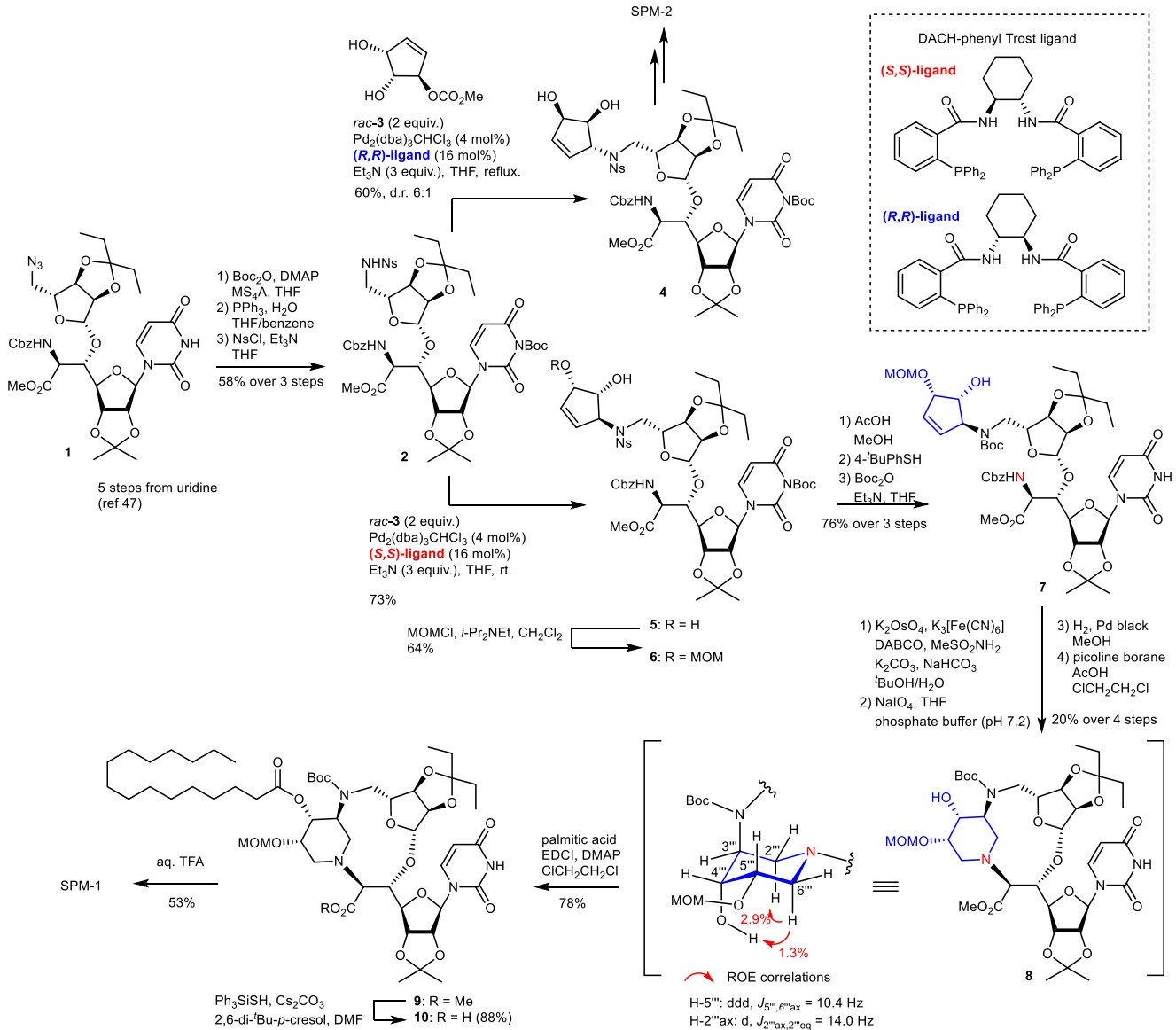

**Fig. 4 | Synthesis of SPM-1 and SPM-2.** DABCO 1,4-diazabicyclo[2.2.2]octane, dba dibenzylideneacetone, EDCI 1-ethyl-3-(3-dimethylaminopropyl)carbodiimide, Ns nosy, ROE rotating frame nuclear Overhauser effect.

spot on MraY. The uridine pocket of MraY is comprised of amino acid residues K70, G194, L195, D196, and N255 (numbering for MraY$_{AA}$) and it is capped by a conserved phenylalanine (F262), which engages in a critical π-π stacking interacting with the uracil moiety. The uracil moiety of SPM-1 makes interactions in this pocket very similar to that of other MraY nucleoside inhibitors (Fig. 5c)[52–54]. Residues T75, N190, D193, and G264 comprise the uridine-adjacent site in MraY. Two moieties of SPM-1, the 5-aminoribosyl and the ester aliphatic tail linkage, form a network of hydrogen bonds with each residue in this pocket. SPM-1 makes a single contact with the TM9b/Loop E site via H325, an interaction which is conserved in all MraY nucleoside inhibitors except capuramycin. The hydrophobic tunnel in MraY, which is thought to recognize the lipid carrier substrate C$_{55}$-P, accommodates the aliphatic palmitoyl tail of SPM-1. The residue which coordinates the Mg$^{2+}$ cofactor, D265 in MraY$_{AA}$, interacts with the macrocyclic structure of SPM-1, where the 5-aminoribosyl links to the piperidine ring system. Therefore SPM-1 binding to MraY likely displaces Mg$^{2+}$. Consistent with this prediction, we did not observe an electron density peak corresponding to Mg$^{2+}$ near D265 in the active site of MraY$_{AA}$.

SPM-1 shares structural similarities with muraymycin D2, carbacaprazamycin, and tunicamycin in addition to the uridine moiety common to all MraY nucleoside inhibitors (Fig. 6). For example, SPM-1, carbacaprazamycin, and tunicamycin each have an aliphatic tail moiety, which localizes to the hydrophobic binding groove in MraY. The positioning of the aliphatic chain in SPM-1 is more similar to tunicamycin than carbacaprazamycin, where it is directed to the hydrophobic groove via interaction between N190 and a linker (an ester in SPM-1 and an amide in tunicamycin) (Fig. 6a). On the other hand, the aliphatic chain in carbacaprazamycin, located away from N190, is also directed to the hydrophobic groove, suggesting there are multiple ways to direct an aliphatic chain to the hydrophobic groove. SPM-1, muraymycin D2, and carbacaprazamycin each have a 5-aminoribosyl moiety, which binds to the uridine-adjacent site on MraY (Fig. 6b). All three of these inhibitors make interactions with T75, D193, and the backbone of G264; however, because in SPM-1, the 5-aminoribosyl moiety is cyclized to the piperidine ring, it is twisted slightly, putting this moiety close enough to the Mg$^{2+}$-coordinating residue, D265, to pick up another interaction (Fig. 6b). D265 also forms a salt bridge interaction with K133, which helps shape the binding pocket for the

**Table 1 | Antibacterial activity of SPMs against ESKAPE**

| | | SPM-1 | SPM-2 | SPM-3 | Sphaerimicin A[b] |
|---|---|---|---|---|---|
| IC$_{50}$ for MraY$_{SA}$ (nM) | | 9.1 | 330 | 41 | – |
| Bacterial spp. | Strains | MIC (µg/mL)[a] | | | |
| *E. faecium* | ATCC 35667 | 4 | 16 | 2 | 2[c] |
| *S. aureus* | ATCC 29213 | 2 | >64 | 8 | 4[d] |
| *K. pneumoniae* | ATCC 13883 | >64 | >64 | >64 | >128 |
| *A. baumannii* | ATCC 19606 | >64 | >64 | >64 | >128 |
| *P. aeruginosa* | ATCC 27853 | >64 | >64 | >64 | >128[e] |
| *E. cloacae* | ATCC 13047 | >64 | >64 | >64 | >128 |
| *E. coli* | ATCC 25922 | >64 | >64 | >64 | >128 |
| *M. tuberculosis* | H37Rv | 16 | >32 | 16 | – |
| *M. avium* | ATCC 25291 | 2 | 16 | 4 | – |

[a]MICs against *M. tuberculosis* and *M. avium* were measured as follows. Bacterial strains were grown at 37 °C for 2 weeks in 200 µL of 7H9 liquid medium containing various concentrations of each compound. MICs were determined as the minimum concentration of compound required to inhibit at least 50% of bacterial growth. The data were obtained from at least two independent biological replicates. MICs against the other bacterial species were determined by a micro-dilution broth method as recommended by the CLSI with cation-adjusted Mueller–Hinton broth (CA-MHB). Twofold serial dilutions of each compound were made in appropriate broth, and the plates were inoculated with 5 × 10⁴ CFU of each strain in a volume of 0.1 mL. Plates were incubated at 37 °C for 20 h and then MICs were scored. The data were obtained from at least three independent biological replicates.
[b]Excerpted from the report[35].
[c]*E. faecium* ATCC 29212.
[d]*S. aureus* ATCC 6538P.
[e]*P. aeruginosa* ATCC 15692.

macrocycle structure of SPM-1. Although the overall conformation of SPM-1 bound to MraY$_{AA}$ is similar to that muraymycin D2 and carba-caprazamycin, it exhibits several distinct features (e.g., aliphatic chain location and the interaction of the 5-aminoribosyl moiety with the enzyme) due to its restricted conformation.

### Inhibition of MraY enzymatic activity by SPM analogues

The inhibitory activity of SPM-1 and SPM-2 was evaluated against MraY$_{AA}$, which was used for our structural studies (Fig. 7). SPM-1 and SPM-2 inhibit MraY$_{AA}$ with IC$_{50}$ values of 0.17 µM and 9.2 µM, respectively using the UMP-Glo assay[55]; SPM-1 is 54-fold more potent than SPM-2 against MraY$_{AA}$ (Fig. 7a). Our structure of MraY$_{AA}$ bound to SPM-1 is consistent with the difference in activity observed between SPM-1 and **S**PM-2. These results clearly indicate that stereochemistry at the piperidine ring greatly affects the global conformation of the macro-cycle of SPMs and that appropriate conformational restriction is important for tight binding to MraY. The inhibitory activity of SPM-1 and SPM-2 was also investigated as against the MraY orthologue from *S. aureus* (MraY$_{SA}$). For the enzymatic reactions catalyzed by MraY$_{SA}$, a fluorescence-based MraY inhibitory assay[56] using dansylated Park's nucleotide[57] was employed (Fig. 7b). SPM-1 and SPM-2 inhibit MraY$_{SA}$ with IC$_{50}$ values of 9.1 nM and 330 nM, respectively; SPM-1 is 36-fold more potent than SPM-2 against MraY$_{SA}$. This is in good accordance with the observed antibacterial activity of these compounds.

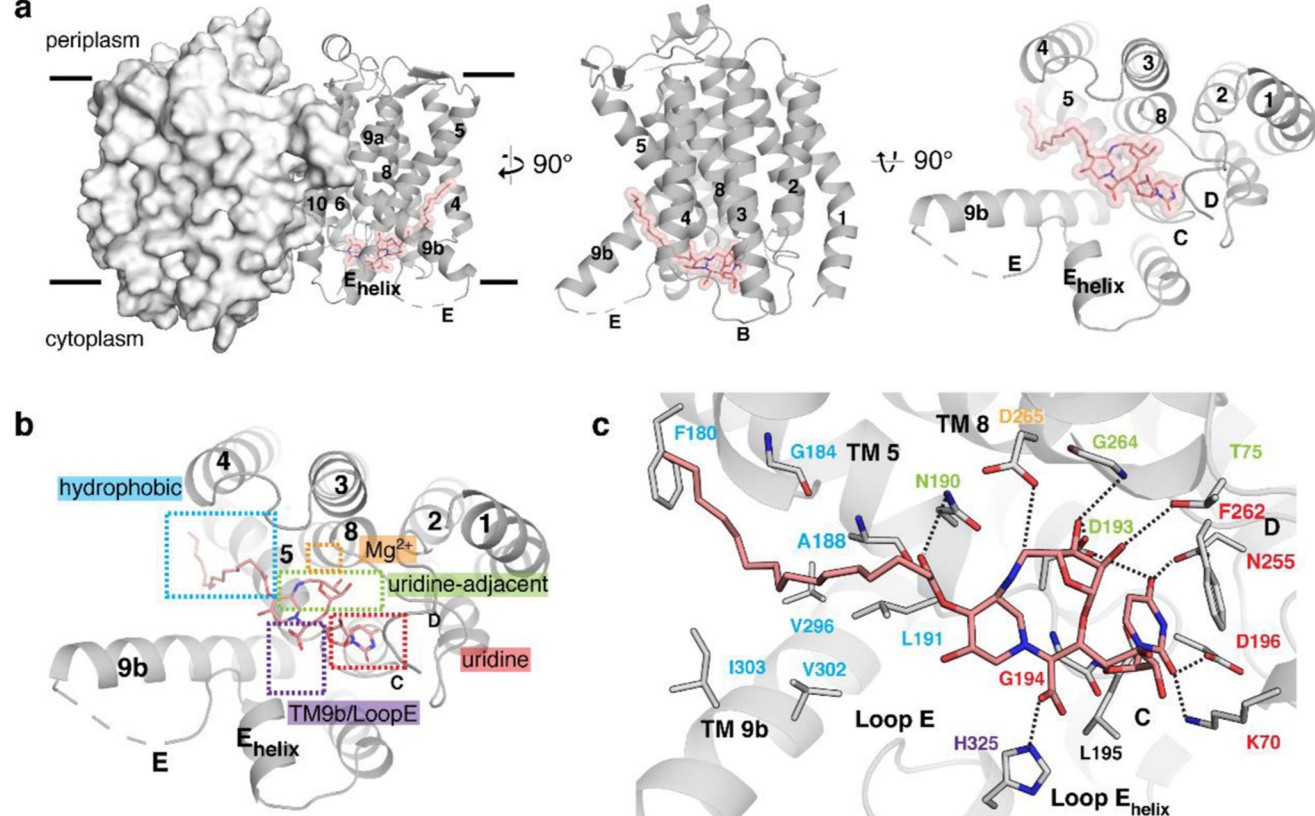

**Fig. 5 | X-ray crystal structure of MraY$_{AA}$ bound to SPM-1. a**, left: The MraY$_{AA}$-SPM-1 complex structure as viewed from the membrane, with one protomer shown in surface representation and one in the cartoon. SPM-1 is shown in salmon. For simplicity, one protomer of MraY$_{AA}$ with bound SPM-1 is shown from the membrane (center) and cytoplasmic views (right). **b** The binding sites recognized by SPM-1 (salmon) on the cytoplasmic side of MraY$_{AA}$ include the uridine (red), uridine-adjacent (lime green), TM9b/Loop E (purple), and hydrophobic (cyan) pockets. **c** A zoomed-in view of the SPM-1 binding site in the same orientation as shown in (**b**). Residues forming interactions with SPM-1 are labeled and color-coded according to the binding pocket to which they belong. Hydrogen bonds are represented by black dashed lines. SPM-1 binds on the cytoplasmic face of MraY formed by TMs 5, 8, and 9b and Loops C, D, and E (labeled throughout).

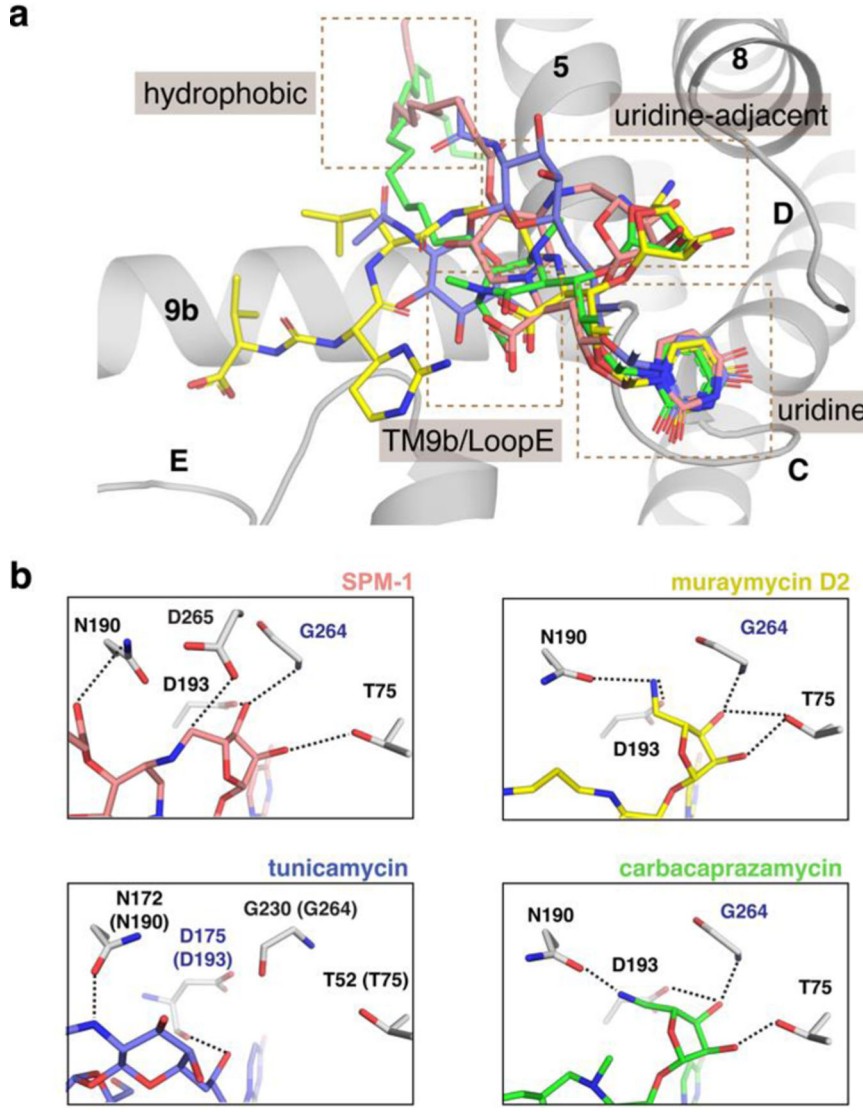

**Fig. 6 | Structural comparison of MraY bound to SPM-1, muraymycin D2, car-bacaprazamycin, and tunicamycin. a** Structural superimposition of SPM-1-MraY$_{AA}$, muraymycin D2-MraY$_{AA}$, carbacaprazamycin-MraY$_{AA}$, and tunicamycin-MraY$_{CB}$ complexes. The binding sites recognized by each inhibitor on the cyto-plasmic side of MraY$_{AA}$ include the uridine, uridine-adjacent, TM9b/Loop E), and hydrophobic pockets. **b** SPM-1, muraymycin D2, tunicamycin, and carbacapraza-mycin binding interactions at the uridine-adjacent site. SPM-1 forms an additional hydrogen bond with the residue residue for coordinating the Mg$^{2+}$ cofactor, D265. SPM-1 is shown in salmon, muraymycin D2 is shown in orange, tunicamycin is shown in slate, and carbacaprazamycin is shown in magenta throughout.

## Structure-based design, synthesis, and biological evaluation of SPM-3

Identification that SPM-1 retained potent biological activity (Table 1 and Fig. 7) as well as elucidation of the molecular interactions of SPM-1 bound to MraY$_{AA}$ (Fig. 5) set the stage for applying structure-based drug design. A difference in the position of the lipophilic side chain is observed when comparing to the complex structures of SPM-1 and carbacaprazamycin bound to MraY$_{AA}$, respectively (Fig. 6). Carbaca-prazamycin possesses potent MraY$_{SA}$ inhibitory activity (IC$_{50}$ 3.8 nM) and antibacterial activity (MIC 8 µg/mL for *S. aureus* ATCC 29213)[58], and these biological activities are similar to those of SPM-1 (Table 1). We sought to transpose the palmitoyloxy group of SPM-1 to the adja-cent 5‴-hydroxyl group on the piperdine. This modification was pre-dicted to be well-tolerated because the 5‴-hydroxyl group in SPM-1 is exposed to solvent and does not interact with MraY$_{AA}$ (Fig. 5). While the resulting 4‴-hydroxy group could potentially be used for future optimization, it was deleted in this study for the following reasons. SPM-1 has many polar functional groups. Decreasing the polarity of a compound increases its bacterial cell membrane permeability,

thus leading to increased antimicrobial activity. The deletion of the 4‴-hydroxy group is expected to reduce the polarity of the molecule but not significantly alter the global conformation. In addition, this analogue is much more accessible from a chemical synthetic point of view. These ideas form the basis of the design of SPM-3 (Fig. 8), which was synthesized in a manner similar to that of SPM-1 (Supplementary Fig. 15). The truncation of the 4‴-hydroxy group also improved the chemical yield of the macrocycle construction (48% over four steps) compared to the synthesis of SPM-1 and SPM-2.

SPM-3 inhibited MraY$_{SA}$ with an IC$_{50}$ value of 41 nM, which is 4.5-fold weaker compared to SPM-1 but still retains MraY$_{SA}$ inhibitory activity (Fig. 7b), serving to validate our design strategy. The observed decrease in apparent affinity is likely due to the fact that SPM-3 lacks the 4‴-hydroxy group that would interact with N190 in MraY. The anti-bacterial activity of SPM-3 was then examined against ESKAPE patho-gens (Table 1). SPM-3 demonstrated antibacterial activity against *S. aureus* and *E. faecium* with MIC values of 8 and 2 µg/mL, respectively. SPM-3 exhibited a similar potency to SPM-1 against *S. aureus* and *E. faecium*. The decreased activity toward MraY inhibition from the loss of

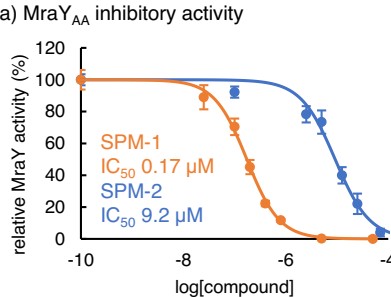

a) MraY$_{AA}$ inhibitory activity

SPM-1
IC$_{50}$ 0.17 µM
SPM-2
IC$_{50}$ 9.2 µM

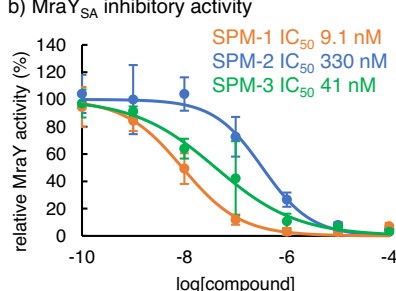

b) MraY$_{SA}$ inhibitory activity

SPM-1 IC$_{50}$ 9.1 nM
SPM-2 IC$_{50}$ 330 nM
SPM-3 IC$_{50}$ 41 nM

**Fig. 7 | MraY inhibitory activity of SPMs. a** The inhibitory activities of the compounds against purified MraY$_{AA}$. Assays were conducted with 100 mM Tris-HCl, 500 mM NaCl, 10 mM MgCl$_2$, and 20 mM (3-((3-cholamidopropyl) dimethyl-lammonio)-1-propanesulfonate), 150 µM UDP-MurNAc-pentapeptide, 250 µM C$_{55}$-P, and MraY$_{AA}$ (50 nM). After 5 min of incubation at 45 °C, the formation of UMP was monitored by luminescence measurement. **b** The inhibitory activities of the compounds against purified MraY$_{SA}$. Assays were conducted with 50 mM Tris-HCl

(pH 7.6), 50 mM KCl, 25 mM MgCl$_2$, 0.2% Triton X-100, 8% glycerol, 50 µM C$_{55}$-P, 10 µM UDP-MurNAc-dansylpentapeptide, and MraY enzyme (11 µg/mL, 5 µL/well). After 3–4 h incubation at room temperature, the formation of dansylated lipid I was monitored by fluorescence enhancement (excitation at 355 nm, emission at 535 nm). The data were obtained from three independent experiments and represented as mean ± SD. SPM-1 is shown in orange, SPM-2 is shown in blue, and SPM-3 is shown in green throughout.

**Fig. 8 | Structure-based design of SPM-3.** The palmitoyl group of SPM-1 is moved to the adjacent 5′′′-hydroxyl group. The resulting 4′′′-hydroxy group is truncated to reduce the polarity of the molecule.

a hydrogen bond with N190 was presumably compensated by reduction of its hydrophilicity. These results prompted us to investigate its antibacterial efficacy against drug-resistant and clinically isolated strains. As shown in Table 2, SPM-3 shows moderate antibacterial activity against methicillin-resistant *S. aureus* (MRSA) JE2 strain. This is also effective against clinically isolated MRSA. They are more effective against a range of vancomycin-resistant Enterococci (VRE) such as *E. faecium* ATCC 51559, 51858 and *E. faecalis* 51299 as well as clinically isolated *E. faecium* and *E. faecalis* with the MIC values ranging from 0.5 to 8 µg/mL. Overall, SPM-3 exhibits preferable activity against Enterococci and *S. aureus*, respectively, although their differences are subtle compared to SPM-1. Most of the clinically isolated MRSA and *E. faecium* tested in this study have additional resistance to levofloxacin, which is a clinically used broad-spectrum antibacterial drug that inhibits DNA gyrase and topoisomerase. It is noteworthy to mention that SPM-3 exhibited potent antibacterial activity against these multidrug-resistant strains. As shown in Table 1, SPM-3 also exhibited better antibacterial activity than SPM-1 against Mycobacterium species including *M. tuberculosis*, which is a causative bacteria of tuberculosis (TB). Being a disease primarily of the respiratory system, TB kills 1.5 million people each year. With resistant strains continuing to emerge, the need for better anti-TB agents possessing new mechanisms of action remains critical. These results suggest that MraY is an attractive target, and its inhibitors are a promising lead for drug-resistant bacterial drugs.

In conclusion, we have rationally designed and synthesized the simplified sphaerimicin analogues, SPMs. As the first step, we predicted a stereoisomer of a sphaerimicin skeleton with potent MraY inhibitory activity out of the eight possible stereoisomers, assisted by molecular modeling, and then designed SPM-1 and SPM-2. These

analogues were synthesized by using two key reactions, which are the asymmetric Tsuji–Trost allylic alkylation reaction and the double-reductive amination to construct the characteristic macrocycle. This is the first report to construct the core skeleton of sphaerimicin, which could be applicable to the synthesis of other diastereomers at the piperidine ring. By evaluating the biological activity of SPM-1 and SPM-2 we found that SPM-1 possesses MraY-inhibiting and antibacterial activity comparable to sphaerimicins. We have also successfully solved the three-dimensional structure of one of these analogues, SPM-1, bound to MraY$_{AA}$. Our structural and biological analyses reveal that the stereochemistry of the macrocyclic core of the SPMs is critical for inhibitor potency. The insight obtained from these studies allowed us to execute the second step drug design of SPM-3, which is chemically more accessible while retaining similar potency. The analogue SPM-3 exhibited potent antibacterial activity against Gram-positive bacteria, including drug-resistant strains such as MRSA and VRE, and clinically isolated multidrug-resistant *S. aureus* and *E. faecium* strains. Notably, there is ample room to optimize SPM-3 to increase its potency and efficacy; for example, by introducing a functional group at the 4′′′-hydroxyl group to improve its interactions with MraY or replace the palmitoyl group with different hydrophobic groups to increase potency against MraY and enhance its delivery across the membrane. Last, our derivatives may be used together with other antibiotics to enhance their efficacy on Gram-negative bacteria.

Generally, it is important to understand and control the protein-bound ligand conformation. Sphaerimicin analogues are highly conformationally restricted due to their unique macrocyclic structure, enabling prediction of their conformation. The macrocyclic skeleton found in this study and its detailed molecular interaction with MraY

**Table 2 | Antibacterial activity of SPM-3 against drug-resistant and clinically isolated strains**

| Bacterial spp. | Strains | MIC (µg/mL)[a] | | | |
|---|---|---|---|---|---|
| | | SPM-3 | Vancomycin | Ampicillin | Levofloxacin |
| *S. aureus*[b] | ATCC 29213 | 8 | 1 | 1 | 0.13 |
| *S. aureus* (MRSA) | JE2 | 8 | 1 | 4 | 8 |
| Clinically isolated MRSA (10)[c] | 4–>64 | 1–2 | 16–32 | 0.13–>32 | |
| *E. faecium*[b] | ATCC 35667 | 2 | 0.5 | 1 | 4 |
| *E. faecium* (VRE) | ATCC 51559 | 2 | >128 | 128 | 16 |
| *E. faecium* (VRE) | ATCC 51858 | 8 | 128 | 128 | 4 |
| *E. faecalis* (VRE) | ATCC 51299 | 1 | 128 | 1 | 1 |
| Clinically isolated *E. faecium* (14)[c] | | 0.5–4 | 0.5–1 | 0.5–>128 | 0.5–>32 |
| Clinically isolated *E. faecalis* (15)[c] | | 0.5–1 | 1-4 | 0.5–1 | 0.5–2 |

[a]MICs were determined by a microdilution broth method as recommended by the CLSI with cation-adjusted Mueller–Hinton broth (CA-MHB). Twofold serial dilutions of each compound were made in appropriate broth, and the plates were inoculated with $5 \times 10^4$ CFU of each strain in a volume of 0.1 mL. Plates were incubated at 37 °C for 20 h and then MICs were scored. The experiments were performed once.
[b]The data are identical to Table 1.
[c]Numbers in parenthesis indicate the number of the strains evaluated for antibacterial activity.

provides a scaffold for developing potent MraY inhibitors, which could be promising leads for antibiotics against drug-resistant bacteria.

## Methods

### Preparation of SPMs
See Supplementary materials.

### Expression, purification, and crystallization of MraY$_{AA}$ in complex with SPM-1
MraY$_{AA}$ was crystallized bound to SPM-1 with the aid of an MraY$_{AA}$-specific camelid nanobody, NB7. NB7 was previously identified as a potent MraY$_{AA}$ binder that recognizes the periplasmic face of the enzyme, away from the cytoplasmic active site, and therefore does not interfere with MraY$_{AA}$ activity or inhibition[53]. MraY$_{AA}$ and NB7 were expressed and purified as previously described. Briefly, the gene corresponding to MraY$_{AA}$ was synthesized as a fusion with a decahistidine-maltose binding protein (His$_{10x}$-MBP), which was codon optimized for expression in *E. coli*. A PreScission protease site between MraY$_{AA}$ and His-MBP was introduced. MraY$_{AA}$ was expressed at 37 °C for 4 h in C41 (DE3) cells. The His$_{10x}$-MBP-MraY$_{AA}$ fusion protein was extracted with dodecyl-maltoside (DDM, Anatrace) and purified using a Co$^{2+}$ affinity resin (Talon). MraY$_{AA}$ was isolated by cleaving His$_{10x}$-MBP tag using PreScission protease (4 °C, overnight). MraY$_{AA}$ was combined with NB7 at a 1:1.5 molar ratio, and the complex was purified by size-exclusion chromatography (SEC) with a Superdex 200 10/300 GL column in 20 mM Tris-HCl, 150 mM NaCl, and 5 mM decyl-maltoside (DM, Anatrace). The peak fractions containing the MraY$_{AA}$-nanobody complex were harvested, concentrated to ~450 µM, and combined with SPM-1 at 1:1.5 molar ratio of protein to inhibitor. The MraY$_{AA}$-NB7-inhibitor complexes were screened for crystallization using sitting drop vapor diffusion with MemGold™ (Molecular Dimensions) and in-house crystallization screening solutions. Crystals of MraY$_{AA}$-NB7-SPM-1 formed at 17 °C in 18% polyethelyene glycol (PEG) 4000, 0.2 M ammonium thiocyanate, 0.1 M sodium acetate pH 4.6. Crystals were equilibrated to 4 °C for 24 h before harvesting and flash cooling.

### Data collection and structure determination
X-ray crystal diffraction data were collected on the NE-CAT 24-IDC and 24-ID-E beamlines (Advanced Photo Source, Argonne National Laboratory) using a wavelength of 0.979 Å. XDS[59] (Version: January 26, 2018) was used to process datasets from two isomorphous crystals, which were merged using BLEND[60] in the CCP4 software suite (Version 7.0). Phasing was obtained by molecular replacement in PHASER[61] in the CCP4 software suite (Version 7.0) using as a search model the structure of MraY$_{AA}$-NB7-carbacaprazamycin (PDB ID: 6OYH) with the

inhibitor, TM9b, Loop E, and the Loop E helix removed. Crystals were in the P21 space group with four NB7 molecules and two MraY$_{AA}$ dimers and in the asymmetric unit. Inhibitor density was strongest in one MraY$_{AA}$ protomer, probably due to crystal packing facilitated by NB7 binding. Jelly-body refinement was first performed on the initial molecular replacement solution using LORESTR[62] in the CCP4 software suite (Version 7.0). Manual model building was performed in COOT[63] (0.8.9) and refinement in PHENIX.refine (Version 1.14-3260)[64]. Molecular graphics were generated using PyMOL (Version 2.0.7)[65]. Data collection and refinement statistics are provided in Supplementary Table 6.

### UMP-Glo assay
The UMP-Glo™ glycosyltransferase assay[55] was carried out in accordance with the manufacturer's specifications (Promega Corporation). For IC$_{50}$ measurements, the reaction buffer contained 100 mM Tris-HCl, 500 mM NaCl, 10 mM MgCl$_2$, and 20 mM (3-((3-cholamidopropyl) dimethylammonio)-1-propanesulfonate) (CHAPS, Anatrace). Reaction mixtures contained 150 µM UDP-MurNAc-pentapeptide (UM5A) and 250 µM undecaprenyl phosphate (C$_{55}$-P) and were initiated with the addition of MraY$_{AA}$ to a final concentration of 50 nM. Reactions were carried out at 45 °C for 5 min. The following concentrations of SPM-1 and SPM-2 were used. SPM-1: 0, 0.025, 0.1, 0.2, 0.4, 0.8, 5, and 50 µM; SPM-2: 0, 0.1, 2.5, 5, 12.5, 25, 70, and 140 µM. A SpectraMax M3 multi-mode microplate reader was used to make luminescence measurements, which were normalized to a negative control reaction without enzyme.

### Fluorescence-based MraY inhibitory assay[56,57]
Reactions were carried out in a 384-well microplate. A solution containing 10 µM dansylated Park's nucleotide and 50 µM undecaprenyl phosphate (C$_{55}$-P, Larodan) in 20 µL of an assay buffer [50 mM Tris-HCl (pH 7.6), 50 mM KCl, 25 mM MgCl$_2$, 0.2% Triton X-100, 8% glycerol] was prepared. The reaction was initiated by the addition of *S. aureus* MraY enzyme (11 µg/mL, 5 µL/well). After 3 h of incubation at room temperature, the formation of dansylated lipid I was monitored by fluorescence enhancement (excitation at 355 nm, emission at 535 nm) from the Infinite M200 microplate reader (Tecan). The inhibitory effects of each compound were determined in the MraY assays described above ($n = 3$). The mixtures contained 2% DMSO in order to increase the solubility of the compounds. IC$_{50}$ values were calculated using a nonlinear regression curve fit with a variable slope in GraphPad Prism version 4.0a.

### Evaluation of antibacterial activity
MICs against *M. tuberculosis* and *M. avium* were measured as follows. Bacterial strains (starting OD$_{600}$ of 0.01) were grown at 37 °C without

shaking for two weeks in 200 μL of supplemented 7H9 liquid medium (Middlebrook 7H9 medium (Difco) supplemented with 10% (vol/vol) oleic acid-albumin-dextrose-catalase (OADC, Difco), 0.2% (vol/vol) glycerol, and 0.05% (vol/vol) tyloxapol (Sigma)) containing various concentrations of each compound prepared in a 96-well flat-bottom plate (TPP). After 2 weeks of incubation at 37 °C, the $OD_{600}$ was measured by Synergy HTX Multi-Mode Reader (Agilent) or Absorbance 96 (byonoy). MICs against ESKAPE were determined as the minimum concentration of compound required to inhibit at least 50% of bacterial growth relative to growth in the no-compound control culture. The data were obtained from at least two biological replicates. MICs were determined by a microdilution broth method as recommended by the CLSI with cation-adjusted Mueller-Hinton broth (MHB). Serial twofold dilutions of each compound were made in appropriate broth, and the strains were inoculated with $5 \times 10^5$ cfu/mL in 96-well plates (each 0.1 mL/well). The plates were incubated at 37 °C for 18 h, and then MICs were determined.

### Reporting summary

Further information on research design is available in the Nature Portfolio Reporting Summary linked to this article.

## Data availability

Data supporting the findings of this manuscript are available from the corresponding author upon request. The source data underlying atomic coordinates and structure factors for the reported crystal structures are deposited in the Protein Data Bank under accession code 8CXR. The Cartesian coordinates in Fig. 2 are provided in Supplementary Data 1. Source data are provided with this paper.

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

## Acknowledgements

We thank Dr. K. Nishiguchi and Dr. S. Arioka (Shionogi Co., Ltd.) for MraY expression and purification. We thank Dr. Justin Fedor and Ali Hao for critical manuscript reading and help with figures, respectively. This research was supported in part by JSPS KAKENHI Grant-in-Aid for Scientific Research (B) (Grant Number JP16H05097 and JP19H03345 to S.I., JP21H03622 to T.S.), Grant-in Aid for Scientific Research on Innovative Areas "Frontier Research on Chemical Communications" (Nos. JP18H04599 and 20H04757 to S.I.), JSPS KAKENHI Grant-in-Aid for Research for Young Scientist (Grant Number JP19K16648 to T.S.), Takeda Foundation, The Tokyo Biomedical Research Foundation and was partly supported by Hokkaido University, Global Facility Center (GFC), Pharma Science Open Unit (PSOU), funded by MEXT under "Support Program for Implementation of New Equipment Sharing System", Platform Project for Supporting Drug Discovery and Life Science Research (Basis for Supporting Innovative Drug Discovery and Life Science Research (BINDS)) from AMED under Grant Number JP18am0101093j0002 and JP22ama121039 to S.I. and A.K., AMED under Grant Number JP19ak0101118h0001, AMED under Grant Number 21ak0101118h9903 to T.S., JST START Program: ST211004JO to T.S., Japan Initiative for Global Research Network on Infectious Diseases (J-GRID) from the Ministry of Education, Culture, Sport, Science, and Technology in Japan, and MEXT for the Joint Research Program of the Research Center for Zoonosis Control, Hokkaido University, and National Institute of Health (R01GM120594) to S.-Y.L. This work used NE-CAT beamlines (GM124165), a Pilatus detector (RRO29205), an Eiger detector (OD021527) at the APS (DE-AC02-06CH11357).

## Author contributions

T.N., S.Y.L., and S.I. designed the research, and T.N., E.H.M., T.S., K.Y., A.K., S.T., M.H., S.Y., S.Y.L., and S.I. designed the experiments. A.K.

performed calculation. T.N., M.Y., and Y.H. synthesized compounds. E.H.M. crystallized the complex of MraY$_{AA}$ bound to SPM-1 and analyzed the structure. Y.K. performed MraY assay. T.S., M.S., and Y.M. performed in vitro antibacterial assay. T.N., M.Y., E.H.M., T.S., Y.H., K.Y., A.K., S.Y., S.Y.L., and S.I. wrote the paper. All authors discussed the results and commented on the paper and have given approval to the final version of the manuscript.

## Competing interests

The authors declare no competing interests.
