## [Peer Review File · Nature Communications]

Synthesis of macrocyclic nucleoside antibacterials and their interactions with MraYREVIEWER COMMENTS

Reviewer #1 (Remarks to the Author):

The paper by Ichikawa, Lee and colleagues describes the first synthesis of analogues of sphaerimicin nucleoside antibiotics as well as their interaction with the bacterial target protein MraY. The amount of work presented in the manuscript is astonishing and includes the computer-assisted design of the sphaerimicin analogues, their challenging synthesis (due to the macrocyclic core structure), the biological activities of the obtained target compounds and an X-ray crystal structure of an analogue in complex with MraY. Thus, structural insights into target interaction of this class of nucleoside antibiotics are provided for the first time. The experimental work has been very thorough, and the obtained results are well presented and discussed. This is some excellent work on an interesting class of natural products that will surely attract a significant readership. I therefore fully support publication of this fine paper in Nature Communications after a few minor issues have been taken care of.

- 1) Overall, the manuscript (including the Supporting Information) would probably benefit from another round of thorough proofreading and some language polishing.
- 2) Figure 1 contains several errors. First, the glycosidic configuration of lipid II is drawn in an ambiguous way. Second, there is a substituent at one of the sugars missing after transport into the periplasm. Third, the connection of the peptide chains in the schematic representation of the mature peptidoglycan is incorrect. Fourth, the 3'-position in sphaerimicin A is mislabelled as 2'.
- 3) Table 2 could probably be moved into the Supporting Information.
- 4) Why has different assay methodology been used for different orthologues of MraY? This should be explained in the text. The obvious choice would have been to run the assays under identical conditions and just to exchange the protein. Why has this not been done?
- 5) Supporting Information, page S3/S4: There are some obvious mistakes in the description of the pKa prediction. The pKa values of the piperidines were not lower than those of the secondary amines, in contrast to what is written in the text. Also, based on the results, zwitterionic states with piperidine-protonated amines were obviously selected for the conformational analysis, not structures with 3"-protonated amines as stated by the authors (also see Figure 2 for that).
- 6) Supporting Information, page S89: There is a typo in the found mass of compound 10.
- 7) Supporting Information, pages S118 and S123: Why are the protocols for the MraY assays included here? They are already presented in the main paper.

Reviewer #2 (Remarks to the Author):

The topic is relevant and timely, as AMR represent a current and serious and global public threat.

The authors convincingly used a combination of approaches and methodologies to study the interaction of Sphaerimicin with MraY and to rationally design simplified, active analogues.

In my opinion, this work is technically sound in many of its aspects. The authors furnished enough details in the experimental section, experiments are well presented, conclusion adequately supported. Preliminary antibacterial activity was examined against ESKAPE pathogen. The authors should comment on the absence of activity against Gram-negative bacteria. Do the authors have info on the toxicity of the synthesized compounds? and on their stability? Furthermore, the authors should consider to combine these novel compounds with antibiotics already in use, also on Gram-negative bacteria

Reviewer #3 (Remarks to the Author):

The manuscript by Nakaya and co-workers describes the synthesis and evaluation of simplified versions of the natural product sphaerimicin A. The compounds were thoroughly characterized in terms of their antibacterial properties, binding potency to the enzyme MraY and, in one case, its detailed structure in complex with MraY. The end result is a compound with interesting antibacterial properties that warrants further investigation and may serve as a starting point for future drug design efforts to develop a novel antibiotic agent. The results presented are of high importance considering the growth of antimicrobial resistance and the need for drugs with new mechanisms of action.

Overall, the study is well conducted and the results are solid. However, in addition to the minor points listed below, there are three major issues that need to be addressed before I can recommend the manuscript for publication.

The title does not make sense to me. It does not read well and commonly either “in complex with” or “bound to” is used when referring to the structure of a protein with a ligand bound.

As part of the study, the structure of one of the synthesized compounds, SPM-1, bound to the target protein MraY was solved to a resolution of 3.65 Å. This is a rather low-resolution structure which limits the level of detail of protein-ligand interactions that can be deduced. For example, at this resolution it is not possible to determine the position of water molecules that may be important for ligand binding. The manuscript as it is written gives the impression that the structure is of very high quality. Please clarify that this is not the case and that even though the orientation of the compound in the binding pocket is clear, the exact positions of the different atoms are not. Especially the conformation of the fatty acyl tail is difficult to model with certainty.

The inhibitory activity of the SPMs against MraY from two different organisms was measured (presented in Figure 5). The conclusions drawn would be strengthened by including also the reference compound sphaerimicin A in the measurement as a comparison.

Minor comments

Introduction:

p4. “...whose magnitude is as large as or much larger than...” – please remove “much”

p5. “...several classes of nucleoside natural product nucleoside inhibitors” – repetition of nucleoside

p8. please define SPM-Ps

Results and discussion:

p10. “The rationale for this simplification was inspired by the observed improvement in antibacterial activity of the structurally related liposidomycins...” – add reference

p10. “The molecular design reduces the number of stereogenic centers and molecular weight from...” – give values; from XX to YY

p18. Please clarify the reference where it was shown that activity and inhibition is unaltered in the presence of NB7.

p18. A resolution of 3.65 Å is limiting and does not provide a very high quality electron density map, please rephrase.

p20. “...has been subdivided into six hot spot...” - the correct reference is ref 53 (Mashalidis et al Nat Commun 2019)

p20. “...inhibitors the same site...” – incomplete sentence, add “interact with” or similar

p21. "...very similar to that of other MraY nucleoside inhibitors" – add reference to refs 18 and 53

p21. Please clarify that the ligand likely displaces the Mg²⁺ cofactor.

p24. How was the inhibitory activity of SPM-1 and 2 against MraYAA evaluated?

p27. "The antibacterial activity of SPM-3 was then examined against ESKAPE pathogens" – please point to Table 1.

Table 1. It would be useful if values for sphaerimicin A from literature were listed in the table for comparison and not just mentioned in the text as comparable.

Figure 4. The colors of the different inhibitors are similar and difficult to distinguish from each other.

Conclusions:

p24. "These are not independent, but rather interact with each other..." – perhaps "influence" is a better choice of word?

Methods:

p34. How was the *S. aureus* MraY enzyme prepared?

Supplementary material:

In the supplementary material sections 6, 7 and 8 are identical to corresponding method sections in the main paper and should be removed.

Thank you for reviewing our manuscript entitled, ‘**Sphaerimicin Analogues: Design, Synthesis, Biological Evaluation and Elucidating Complex Structure Bound to MraY**’, which we are submitting to *Nature Communications* for publication as *Articles*. We also appreciate the time and effort the reviewers have dedicated to providing insightful feedback on ways to strengthen our paper. Thus, it is with great pleasure that we resubmit our revised version of the manuscript for further consideration. We also hope that our edits and the responses provided below satisfactorily address all the issues and concerns the reviewers have noted.

The following is a point-by-point response:

Reviewers’ comment

1. **Reviewer 1:** Overall, the manuscript (including the Supporting Information) would probably benefit from another round of thorough proofreading and some language polishing.

Per reviewer’s suggestion, we have done additional rounds of proofreading.

2. **Reviewer 1:** Figure 1 contains several errors. First, the glycosidic configuration of lipid II is drawn in an ambiguous way. Second, there is a substituent at one of the sugars missing after transport into the periplasm. Third, the connection of the peptide chains in the schematic representation of the mature peptidoglycan is incorrect. Fourth, the 3'-position in sphaerimicin A is mislabelled as 2'.

Response: We thank the reviewer for catching these errors. We fixed all errors pointed out in Figure 1.

3. **Reviewer 1:** Table 2 could probably be moved into the Supporting Information.

RESPONSE: Table 2 was moved into Supplementally Information as Table S6.

4. **Reviewer 1:** Why has different assay methodology been used for different orthologues of MraY? This should be explained in the text. The obvious choice would have been to run the assays under identical conditions and just to exchange the protein. Why has this not been done?

RESPONSE: We agree that the assays under identical condition would be ideal. In this study, these two assays were conducted by two laboratories for two different sets of MraY from different species. For MraY_{AA} responses against two different SPM analogues (SPM-P1 and SPM-P2 which exhibit different stereo chemistries at the 3''

position), the UMP-Glo assay was used and conducted in the Lee lab, for *MraY* from other pathogenic bacteria, the fluorescent Park nucleotide assay was used in the Ichikawa lab. Although it is ideal to use the same assay to perform the same enzymatic assay for *MraY* from all the species tested in our manuscript, we ensure that the use of two different assays does not affect the interpretation and conclusion of our studies. First, it is reported that results from the fluorescent Park nucleotide assay are not too different from those using the UMP-Glo kit experiment. [Stachyra, T. *et al. Antimicrob. Agents Chemother.* **2004**, *48*, 897. Fluorescence Detection-Based Functional Assay for High-Throughput Screening for *MraY*., Das, D. *et al. Sci. Rep.* **2016** *6*, 33412. Rapid and efficient luminescence-based method for assaying phosphoglycosyltransferase enzymes.] Second, these two assays were used against two different sets of *MraY* orthologues with distinct goals. 1) the UMP-Glo assay was used to compare the stereochemistry of sphaerimicin against *MraY*_{AA} to test which stereochemistry is optimal between SPM-1 and SPM-2) The fluorescent assay was mainly used to test the efficacies of our SPM-3 analogue against *MraY* from pathogenic bacteria. Third, the differences in the IC₅₀ values between SPM-1 and SPM-2 are similar in *MraY*_{AA} and *MraY*_{SA} (36-54 folds higher for the SPM-2), further supporting the use of two different assays. We have revised the manuscript to include the following sentence to clarify the use of two different assays.

p.24 “**SPM-1** and **SPM-2** inhibit *MraY*_{AA} with IC₅₀ values of 0.17 μM and 9.2 μM, respectively using the UMP-Glo assay (reference #65); **SPM-1** is 54-fold more potent than **SPM-2** (Figure 5a).”

5. **Reviewer 1:** Supporting Information, page S3/S4: There are some obvious mistakes in the description of the pK_a prediction. The pK_a values of the piperidines were not lower than those of the secondary amines, in contrast to what is written in the text. Also, based on the results, zwitterionic states with piperidine-protonated amines were obviously selected for the conformational analysis, not structures with 3^{'''}-protonated amines as stated by the authors (also see Figure 2 for that).

RESPONSE: We thank the reviewer for catching this error. We changed “the pK_a values of the piperidines were *lower* than secondary amines” to “the pK_a values of the piperidines were *higher* than secondary amines”. The represented pK_a values in Table S1 are that of that of corresponding protonate states. Thus, the piperidines which show more higher pK_a values are more basic than the secondary amines. The result of the calculation is consistent with our selection of piperidine-protonated twitter ions for further conformational studies.

6. **Reviewer 1:** Supporting Information, page S89: There is a typo in the found mass of compound 10.

RESPONSE: We thank the reviewer for catching this error. The observed m/z was changed to 1037.5889.

7. **Reviewer 1:** Supporting Information, pages S118 and S123: Why are the protocols for the MraY assays included here? They are already presented in the main paper.

RESPONSE: The protocols were removed from the supplementary material sections.

8. **Reviewer 2:** The authors should comment on the absence of activity against Gram-negative bacteria. Do the authors have info on the toxicity of the synthesized compounds? and on their stability?

RESPONSE: The sentence “SPMs could not penetrate into the cells owing to the outer membrane barrier, so has no activity against Gram-negative bacteria.” was added. (p13, line 9)

We do not have adequate information about the toxicity and stability of the synthesized compounds. These important properties would be revealed by upcoming studies.

9. **Reviewer 2:** Furthermore, the authors should consider to combine these novel compounds with antibiotics already in use, also on Gram-negative bacteria.

RESPONSE: The reviewer provided critical comment on the combination of several antibiotics. Several research demonstrated the effectiveness of the combination for the treatment of infection caused by AMR. [one example: Brady, S. F. *et al. Nat. Chem. Biol.* **2016**, *12*, 1004. Discovery of MRSA active antibiotics using primary sequence from the human microbiome.] We are also interested in the activities of our sphaerimicin derivatives in the presence of other antibiotics. We have not yet tested this combination, but plan to conduct as the next stage of our project. We thus included this possibility in Conclusion section. “Last, although our sphaerimicin derivatives do not have efficacy against Gram-negative bacteria, our derivatives may be used together with other antibiotics to enhance their efficacy on Gram-negative bacteria (references – Brady...).”

10. **Reviewer 3:** The title does not make sense to me. It does not read well and commonly either “in complex with” or “bound to” is used when referring to the structure of a protein with a ligand bound.

RESPONSE: The title was changed to “Synthesis of macrocyclic nucleoside antibacterials and their interactions with MraY”

11. **Reviewer 3:** As part of the study, the structure of one of the synthesized compounds, SPM-1, bound to the target protein MraY was solved to a resolution of 3.65 Å. This is a rather low-resolution structure which limits the level of detail of protein-ligand interactions that can be deduced. For example, at this resolution it is not possible to determine the position of water molecules that may be important for ligand binding. The manuscript as it is written gives the impression that the structure is of very high quality. Please clarify that this is not the case and that even though the orientation of the compound in the binding pocket is clear, the exact positions of the different atoms are not. Especially the conformation of the fatty acyl tail is difficult to model with certainty.

Response: We thank the reviewer for bringing up this point. It was not our intention to claim that our structure is of very high quality to precisely assign the entire ligand and the associated water molecules at the atomic level. For that we apologize. In our revision, we have clarified this point as follows in page 15.

“Good-quality electron density maps allowed for the placement of **SPM-1** (Figure S13). Although the orientation of the macrocyclic nucleoside structure is clear, due to the limited resolution, assignment of water molecules bound to the ligand is impossible and the exact atomic positions of the aliphatic chain are unclear.”

12. **Reviewer 3:** The inhibitory activity of the SPMs against MraY from two different organisms was measured (presented in Figure 5). The conclusions drawn would be strengthened by including also the reference compound sphaerimicin A in the measurement as a comparison.

RESPONSE: We think this comment is quite important as the validity of the derivative could be assessed by direct comparison of the activity. However, this natural product has only been isolated in small amount and was not available for our study. Hence, we added reported data for natural sphaerimicin A by Van Lanen in Table 1.

13. **Reviewer 3:** p4. "...whose magnitude is as large as or much larger than..." – please remove "much"
RESPONSE: We removed "much" in the sentence.
14. **Reviewer 3:** p5. "...several classes of nucleoside natural product nucleoside inhibitors" – repetition of nucleoside
RESPONSE: The sentence was fixed to "several classes of nucleoside natural product inhibitors"
15. **Reviewer 3:** p8. please define SPM-Ps
RESPONSE: In the initial version of the manuscript, there were several notations of SPM-Ps, which could cause confusion for readers. We change all "SPM-Ps" to "SPMs" to avoid the confusion.
16. **Reviewer 3:** p10. "The rationale for this simplification was inspired by the observed improvement in antibacterial activity of the structurally related liposidomycins..." – add reference
RESPONSE: References 25 and 39 were added to the end of the sentence.
17. **Reviewer 3:** p10. "The molecular design reduces the number of stereogenic centers and molecular weight from..." – give values; from XX to YY
RESPONSE: We changed the sentence to "The molecular design removes six stereogenic centers and reduces molecular weight from 974 to 784, comparing to the original chemical structure of sphaerimicin A."
18. **Reviewer 3:** p18. Please clarify the reference where it was shown that activity and inhibition is unaltered in the presence of NB7.
RESPONSE: We thank the reviewer for catching this point. We have included the reference (reference #53).
19. **Reviewer 3:** p18. A resolution of 3.65 Å is limiting and does not provide a very highquality electron density map, please rephrase.
RESPONSE: As responded above, we change the sentence as follows in p.18.
"Good-quality electron density maps allowed for the placement of **SPM-1** (Figure S13). Although the orientation of the macrocyclic nucleoside structure is clear, due to

the limited resolution, assignment of water molecules bound to the ligand is impossible and the exact atomic positions of the aliphatic chain are unclear.”

20. **Reviewer 3:** p20. “...has been subdivided into six hot spot...” - the correct reference is ref 53 (Mashalidis et al Nat Commun 2019)

RESPONSE: We thank the reviewer for catching this referencing error. The reference number was fixed to 53.

21. **Reviewer 3:** p20. “...inhibitors the same site...” – incomplete sentence, add “interact with” or similar

RESPONSE: We thank the reviewer to catching this error. Words “interact with” was added to the sentence.

22. **Reviewer 3:** p21. “...very similar to that of other *MraY* nucleoside inhibitors” – add reference to refs 18 and 53

RESPONSE: We thank the reviewer for catching this referencing error. References 18 and 53 were added to the end of the sentence.

23. **Reviewer 3:** p21. Please clarify that the ligand likely displaces the Mg^{2+} cofactor.

RESPONSE: We thank the reviewer for catching this detailed but important point in SPM binding. We included the following sentence in p.22. “Therefore SPM-P1 binding to *MraY* likely displaces Mg^{2+} . Consistent with this prediction, we did not observe an electron density peak corresponding to Mg^{2+} near D265 in the active site of *MraY*_{AA}.”

24. **Reviewer 3:** p24. How was the inhibitory activity of SPM-1 and 2 against *MraY*_{AA} evaluated?

RESPONSE: The inhibitory activity of SPM-1 and SPM-2 against *MraY*_{AA} was evaluated by using the UMP-Glo assay. We used this method for our previous study of *MraY*_{AA} (Mashalidis et al, Nat. Commun. 2019). The detailed method was included in the method section.

25. **Reviewer 3:** p27. “The antibacterial activity of SPM-3 was then examined against ESKAPE pathogens” – please point to Table 1.

Response: We added “(Table 1)” to the end of the sentence.

26. **Reviewer 3:** Table 1. It would be useful if values for sphaerimicin A from literature were listed in the table for comparison and not just mentioned in the text as comparable.

RESPONSE: MIC values of sphaerimicin A from literature were listed in Table 1.

27. **Reviewer 3:** Figure 4. The colors of the different inhibitors are similar and difficult to distinguish from each other.

Response: We have now changed the colors of different inhibitors in the Figure 4 to be able to distinguish the feature of each inhibitor better.

28. **Reviewer 3:** p24. “These are not independent, but rather interact with each other...” – perhaps “influence” is a better choice of word?

RESPONSE: Word “interact” was changed to “influence”.

29. **Reviewer 3:** p34. How was the *S. aureus* MraY enzyme prepared?

RESPONSE: The MraY enzyme was expressed in *E.coli* cells and purified using His-Trap HP column. The details are described in ref 66.

30. **Reviewer 3:** In the supplementary material sections 6, 7 and 8 are identical to corresponding method sections in the main paper and should be removed.

RESPONSE: The overlapping parts of sections 6, 7, and 8 were removed from the supplementary material sections.

Once again, we appreciate all efforts by the reviewers.

REVIEWERS' COMMENTS

Reviewer #1 (Remarks to the Author):

For the revised version of this paper by Ichikawa, Lee and colleagues, the authors have invested quite some work into improvements of their original manuscript based on the reviewers' comments. All concerns have been reasonably addressed. I therefore vote for publication of this very fine paper in Nature Communications.

Reviewer #2 (Remarks to the Author):

The authors adequately answered to my comments

Reviewer #3 (Remarks to the Author):

I am happy with the revised version of the paper and the response to my questions by the authors. I therefor support publication of the manuscript.